# Stimulation of B-Lymphopoiesis by Administration of a Trimecaine-Based Ionic Compound in Cyclophosphamide-Induced Hematopoietic-Depressive Model

**DOI:** 10.3390/molecules28031378

**Published:** 2023-02-01

**Authors:** Layilya Baktybayeva, Guldana Daulet, Alexey Zazybin, Valentina Yu, Yekaterina Ostapchuk, Yuliya Perfilyeva, Aikyn Kali, Nurshat Abdolla, Aigul Malmakova, Nuraly Baktybai, Zhanerke Temirbekova, Khadichahan Rafikova

**Affiliations:** 1Department of Biophysics, Biomedicine and Neuroscience, Al-Farabi Kazakh National University, Al-Farabi Av., 71, Almaty 050040, Kazakhstan; 2School of Chemical Engineering, Kazakh British Technical University, Tole Bi Str., 59, Almaty 050000, Kazakhstan; 3Laboratory of Synthetic and Natural Medicinal Compounds Chemistry, A.B. Bekturov Institute of Chemical Sciences, Walikhanov Str., 106, Almaty 050010, Kazakhstan; 4Laboratory of Molecular Immunology and Immunobiotechnology, M.A. Aitkhozhin’s Institute of Molecular Biology and Biochemistry, Dosmukhamedov Str., 86, Almaty 050012, Kazakhstan; 5Department of Chemical and Biochemical Engineering, Institute of Oil and Gas Geology, Satbayev University, Almaty 050013, Kazakhstan

**Keywords:** B-lymphopoiesis, trimecaine-based ionic compound, cyclophosphamide-induced hematopoietic-depressive model

## Abstract

According to the WHO, the secondary form of hematopoietic-depressive status increases the risk of death in people with oncological, infectious, and hormonal diseases. The choice of drugs that stimulate the hematopoietic activity of B-lymphopoiesis is limited. The current leucopoiesis drugs have a number of side effects: thymic preparations stimulate the production of PGE2, which causes chronic inflammation and various autoimmune diseases through the differentiation of T helper 1 (Th1) cells, the proliferation of Th17 cells, and the production of IL-22 from Th22 cells through EP2 and EP4 receptors; cytokine preparations can cause uncontrolled immune reactions and impaired contractility of smooth and cardiac muscles; drugs based on nucleic acids can stimulate the division of all cells, including bacterial and cancerous ones. The use of oligonucleotides such as ribozymes and antisense oligodeoxynucleotides (AS-ODNs) shows promise as therapeutic moieties, but faces a number of challenges such as nuclease sensitivity, off-target effects, and efficient delivery. The search for substances that stimulate B-lymphopoiesis among ionic compounds was motivated by the discovery of the unique properties of lidocaine docusate, one of the first ionic liquid forms of the known drugs. The lidocaine docusate (protonated form of lidocaine (2-(diethylamino)-N-(2,6-dimethylphenyl) acetamide + docusate-anion (dioctylsulfosuccinate))) suppresses the division of pheochromocytoma cells and activates immunity in rats. The trimecaine-based ionic compound (TIC) demonstrates high B-lymphopoiesis-stimulating activity. The TIC compound stimulates an increase in the volume of transitional B cells, which play an important role for further differentiation and formation of a sufficient number of mature B1 cells and mature B2 cells, where mature B2 cells make up the bulk of the functional population of B lymphocytes. The TIC compound most strongly stimulated the restoration of the number of marginal zone B cells, follicular B cells, and activated germinal center B cells after the cytotoxic emptying of the follicular centers of the spleen induced cyclophosphamide. It significantly exceeds the activity of the comparison drug methyluracil. The TIC compound does not affect the level of pro-B, pre-B-I, or pre-B-II bone marrow cells, which prevents the risk of the formation of immature functionally defective cells.

## 1. Introduction

The most common form of acquired B-lymphocyte immunodeficiency is secondary antibody deficiency [1]. Secondary antibody deficiency is associated with protein loss due to diseases of the kidneys, gastrointestinal tract, severe burns, dermatitis, peritoneal dialysis, myotonic dystrophy, and malnutrition. The next most common form of secondary B-lymphocyte immunodeficiency is associated with the suppression of proliferative activity in primary and secondary lymphoid–myeloid organs. The causes are iatrogenic and radiation therapy, transplantation, HIV infection, regulatory disorders of T cells, dendritic cell dysfunction, circadian rhythm disturbance, and hematological diseases [1,2]. The spectrum of clinical consequences of secondary B-lymphocyte deficiency is characterized by sinus–pulmonary recurrent, chronic, systemic, and complicated infections [3]. Infections in these patients are more likely to be caused by encapsulated bacteria (some strains of Haemophilus influenzae), but susceptibility to Clostridium difficile and Escherichia coli, Staphylococcus aureus, and fungal and viral infections is also increased. A study of more than 3000 patients with primary disease multiple myeloma and secondary antibody deficiency found that infection was responsible for 45% of deaths within 6 months of diagnosis. Risk ratios for pneumonia, septicemia, or meningitis were 7.7, 15.6, and 16.6 times higher, respectively, in patients with multiple myeloma and secondary antibody deficiency compared to age-matched controls [1]. There are very few drugs with lymphopoiesis-stimulating activity, and they have side effects. Preparations are based on sodium deoxyribonucleate: sodium nucleate, nucleospermate sodium, pentoxyl, poludan, inosine pranobex, methyluracil, and inosine. The drugs are effective [4], but they contribute to the intensive division of pathogenic microflora and cancer cells, so they are banned for use in many countries of the world. Thymus preparations (timalin, timogen, and tactivin [5]) stimulate the proliferation and differentiation of T-lymphocytes, indirectly stimulating B-lymphopoiesis, which induces the synthesis of interferon, tumor necrosis factors, and colony-stimulating factors [6]. However, thymus preparations stimulate the division of all subpopulations of T-lymphocytes, especially cytotoxic T-lymphocytes, which sometimes causes septic shock. They stimulate the production of PGE_2_ by malignant cells, which leads to pericancrotic inflammation [7] and suppression of antitumor immunity. Cytokine-based preparations contain cytokines, interferons, and colony-stimulating factors [8] (filgrastim, lenograstim, pegfilgrastim, empegfilgrastim, roncoleukin) that stimulate interferogenesis, T- and B-lymphopoiesis [9], and immune responses, and participate in angiogenesis, apoptosis, chemo taxis, and embryogenesis [10,11]. However cytokine drugs can cause uncontrolled immune reactions [12], septic shock, decreased contractility of the myocardium and smooth muscle cells of blood vessels [13], induce “bone” pain, increase permeability of the endothelium, and impair microcirculation in organs, decrease blood pressure [14], and cause hypoglycemia [15]. Preparations from plant extracts are biologically active additives [16,17,18]. Thus, the search for drugs with a directed B-lymphopoiesis-stimulating effect is relevant and necessary.

One of the modern trends in pharmacology is the synthesis of ionic compounds based on known active pharmaceutical ingredients as the cationic part and small ions (halogens, nitrates) as the anionic part [19]. Ionic liquids exhibit high bioavailability, low toxicity, and a long interval of action, and sometimes demonstrate completely new pharmacological activity. For example, ionic liquid lidocaine docusate (protonated form of lidocaine (2-(diethylamino)-N-(2,6-dimethylphenyl) acetamide + docusate-anion (dioctylsulfosuccinate))) suppresses the division of pheochromocytoma cells and activates immunity in rats [20].

Trimecaine was chosen by the authors for its unique properties (local anesthetic and antiarrhythmic agent) and non-toxicity, which are well-known in medicine. At the same time, ionic compounds based on trimecaine have not been studied sufficiently, especially in terms of their pharmacological activity. The first reports on the leucopoiesis-stimulating and plant-growth-regulating activities of the ionic compounds based on trimecaine were published in [21,22,23,24,25,26,27,28,29,30,31].

Thus, the aim of the study was to conduct a study of the affects of trimecaine-based ionic compounds (N,N-diethyl-N-(2-(mesitylamino)-2-oxoethyl) propan-1-aminium iodide) on the stimulating activity of B-lymphopoiesis.

## 2. Results

### 2.1. Chemical Study Results

Cationic part: 2-(Diethylamino)-N-(2,4,6-trimethylphenyl) acetamide. International nonproprietary name: trimecaine. CAS №: 616-68-2. Empirical formula: C_21_H_23_NO_3_·HCl. Anionic part: propane iodide.

Trimecaine-based ionic compound (N,N-diethyl-N-(2-(mesitylamino)-2-oxoethyl) propan-1-aminium iodide) (TIC) was obtained by combining 2-(Diethylamino)-N-(2,4,6-trimethylphenyl) acetamide as the cationic part and propane iodide as the anionic part (Figure 1). Chemical synthesis of the TIC compound is described in an earlier published chemical article [21,23,26]. The B-lymphopoiesis-stimulating activity of the TIC compound (analysis of proliferation pathways of bone marrow and spleen cells) was investigated and presented for the first time in this article. 

### 2.2. Biological Study Results

#### 2.2.1. Analysis of Weight and Cellularity of Bone Marrow and Spleen

The indicators of the average number of the nucleated cells (NC) of the bone marrow in the groups differ significantly. The indicator of the average number of NC cells in the TIC group (17.2 ± 2.4) ·10^6^ cells is lower than in the placebo group (70.9 ± 7.2) (*p* < 0.01) and the methyluracil group (55.4 ± 7.1) by 4.1 and 3.2 times, respectively (*p* < 0.02) (Table 1).

The average weight of the spleen in the TIC group is (45.0 ± 3.3) mg and is less than in the placebo group (60.0 ± 5.0) mg, the methyluracil group (70.0 ± 5.0) mg, and the untreated group (72.5 ± 17.0) mg. The average number of spleen NC cells in the TIC group is (73.7 ± 5.4) ·10^6^ cells, and is slightly higher than in the placebo group (66.4 ± 5.9), and the methyluracil group (60.6 ± 2.6). The average number of spleen NC cells in the TIC group approaches the average value of spleen NC cells of the untreated group (92.1 ± 5.9) ·10^6^ cells (Table 1).

#### 2.2.2. Results of Flow Cytometric Phenotypic Analysis 

##### Analysis of the Hematopoiesis-Stimulating Activity of the TIC Compound on Level Bone Marrow Cells Recovery after Cyclophosphamide-Induced Myelosuppression

Characterization of B cells begins with the stage of differentiation of pro-B cells by phenotype B220**^+^**/CD45R**^+^**CD19**^+^**CD43**^+^**. At the early stage of proliferation and differentiation, the average value of pro-B cells in the TIC group is (4.9 ± 0.0)% and is slightly higher than in the methyluracil group (3.4 ± 0.1)% (*p* < 0.00002, F > F_crit_, 91.21 > 4.96), and in the placebo group (3.5 ± 0.3)% (*p* < 0.00002, F > F_crit_, 30.00 > 4.96), but is 1.5 times lower than in the untreated group (7.6 ± 0.8)% (*p* < 0.00002, F > F_crit_, 52.48 > 4.96) (Figure 2a). 

We combined pre-B cells and immature B bells by phenotype B220/CD45R^+^CD19^+^CD43^−^. In the placebo, methyluracil, and TIC groups, the average value of pre-B cells and immature B cells is in the range (7.6–9.6)%. The average value in the TIC group is 1.8 time lower than the average value of the untreated group (14.0 ± 6.1)% (*p* < 0.0001, F > F_crit_, 33.64 > 4.96) (Figure 2b).

##### Analysis of the Hematopoiesis-Stimulating Activity of the TIC Compound on Level Spleen Cells Recovery after Cyclophosphamide-Induced Myelosuppression


*A—The restoration of immature B-cells and transitional B-cells.*


We combined immature B cells and transitional B cells according to the B220/CD45R+CD19+CD43-IgM+ phenotype. The average level of immature B cells and transitional B cells in the TIC group is (5.4 ± 1.2)% and exceeds the average level of cells in the methyluracil group (2.2 ± 0.2)% (*p* < 0.00006, F > F_crit_, 43.43 > 4.96), and the average level of cells in the untreated group (1.3 ± 0.2)% (*p* < 0.00006, F > F_crit_, 73.02 > 4.96). The average level of immature B cells and transitional B cells is minimal in the placebo group (0.8 ± 0.0)% and is significantly lower than the TIC group by 6.75 times (*p* < 0.00006, F > F_crit_, 100.83 > 4.96) (Figure 2c).


*B—The restoration of follicular B cells and marginal zone B cells.*


Transitional B cells give rise to populations of mature naïve B cells, which form follicular B cells (FO-B cells) and marginal zone B cells (MZB). We combined MZB cells and FO-B cells with a common phenotype B220^+^/CD45R^+^CD19^mid^CD43^−^. At this stage of differentiation, mature naïve B cells are characterized by high proliferative activity. The TIC compound stimulates the division of MZB cells and FO-B cells. The average level of cells in the TIC group (34.4 ± 9.1)% exceeds the average level of cells in the methyluracil group (23.9 ± 4.8)% by 1.4 times (*p* < 0.00004, F > F_crit_, 47.53 > 4.96), and the average level of cells in the placebo group (16.8 ± 3.1)% by 2.0 times (*p* < 0.000002, F > F_crit_, 151.40 > 4.96) (Figure 3b,c). The average volume of MZB cells and FO-B cells of the TIC group (34.4 ± 1.7)% is almost identical to the average level of cells of the untreated group (32.6 ± 3.8)% (Figure 3c).


*C—The restoration of activated germinal center B cells.*


Mature naive B cells express BAFF/APRIL receptors (a proliferation-inducing ligand), connect to an antigen, and are activated. In addition, B cells interact with T lymphocytes, express MHC II^+^ molecules and CD40^+^ co-stimulating molecules, and form a population of activated germinal center B cells. Thus, we registered activated germinal center B cells by phenotype B220^+^/CD45R^+^ CD19^+^ CD40^+^ MHC class II^+^. The average level of activated germinal center B cells reaches (68.8 ± 2.7)% in the TIC group, which is slightly higher than the average level of cells in the untreated group (60.6 ± 2.4)% (*p* < 0.01, F > F_crit_, 8.53 > 4.96) (Figure 4a). The average level of cells in the TIC group (68.8 ± 2.7)% is 4.5 times higher than the average level of cells in the placebo group (15.2 ± 1.5)% (*p* < 0.000006, F > F_crit_, 510.13 > 4.96), and is higher than the average level of cells in the untreated group (36.7 ± 5.3)% by 1.8 times (*p* < 0.01, F > F_crit_, 8.53 > 4.96) (Figure 4b,c). 

## 3. Discussion

The newly synthesized compound TIC refers to ionic compounds. Ionic liquids are usually characterized by high bioavailability, low toxicity, and a long duration of action. These pharmacodynamic properties are a priority in the development of new drugs. We studied the hematopoiesis-stimulating activity of the ionic compound TIC on the cyclophosphamide-induced hematopoietic-depressive model. The cyclophosphamide model has been successfully applied in pharmacological experiments on mice [32]. 

The reference drug was methyluracil. The first reason for the choice is that methyluracil stimulates the recovery of B-lymphocytes [13]. Peptide preparations based on thymus hormones (thymosin, thymopoietin, thymulin) [5], and preparations of bone marrow origin (myelopid, leukogen, lenograstim) [33] do not stimulate the recovery of B-lymphoid cells, while herbal preparations are adaptogens [34,35,36,37,38,39]. The second reason for choosing methyluracil is that the ionic compound N,N-dimethyl-N-(2-(methylamino)-2-oxoethyl) propane-1-ammonium iodide (TIC) is structurally more similar to methyluracil (2,4-dioxo-6-methylpyrimidine). Both compounds contain heterocyclic nuclei, with a keto group in their chemical structures.

At the first stage, we studied the effect of the TIC compound on the ability to restore the NC index in the bone marrow and spleen. The levels of NC cells in the TIC group are lower than in the placebo and methyluracil groups. However, the index in the placebo and methyluracil groups are significantly higher than in the untreated group.

It is known that the proliferative activity of erythroblast, megakaryoblast, and myeloblast cells is higher than that of pools of lymphoid cells. The emptying of the bone marrow stimulates the release of hormones, myeloids, and interleukins, which consequently stimulate an increase in the number of erythrocyte, megakaryocytic, and myelocytic cells. The stimulatory effect of the stress factor can be seen in the high index of NC cells in the bone marrow in the experimental groups, and the low index in the untreated group. However, as the phenotyping scores show, a high index of NC cells in the bone marrow does not indicate an effective recovery of B-lymphocyte cells.

High rates of NC cells in the spleen are in all experimental groups. The index of NC cells in the spleen in the placebo group is identical to the index of NC cells in the spleen in the methyluracil group. It is known that when the bone marrow is empty, the spleen begins to perform a compensatory hematopoietic function in the body of an adult human [5,40]. The stress factor in the placebo group stimulates the restoration of hematopoiesis. The level of NC cells in the spleen in the TIC group is higher than in the placebo group and the methyluracil group. This means that the TIC compound shows a stimulating activity to restore the level of cells in the secondary lympho-myeloid organ.

Next, we analyzed the effect of the TIC on the recovery C-kit (CD117**^+^**)**^−^**, Ly6G**^+^**Ly6C**^+−^**, Ter119**^+−^**, CD19**^+−^**, and CD3e**^+^**-expressing bone marrow cells after cyclophosphamide-induced myelosuppression. The TIC shows a high leukopoiesis-stimulating activity to restore the level of Ly6G**^+^**Ly6C**^+−^** and CD19**^+−^**-expressing bone marrow cells after cyclophosphamide-induced myelosuppression.

The main functional load of B lymphocytes is associated with humoral immunity, where plasma cells secrete antibodies that protect the body from antigens. Also, B lymphocytes participate in various types of immune reactions, performing regulatory, adhesive, and receptor functions; they also participate in antibacterial, antiparasitic, antitumor, and antiviral immunity [2]. Consequently, timely recovery of the B lymphocyte population plays a huge role for patients.

We analyzed stimulated and unstimulated proliferative activity in the B cells in the bone marrow and lymphoid–myeloid organ of the spleen. In the bone marrow, the TIC compound does not affect the process of restoring the level of pro-B cells, pre-B-I, and pre-B-II cells, or that of immature B cells. Many authors believe that at the stage of differentiation of pro-B cells, pre-B-I, and pre-B-II cells, it is very important that the cells are not sensitive to the effects of chemical and physical factors [39]. Some compounds at the stage of pre-B-cell transition can initiate damage in the IL-7 receptor subunit, and they significantly increase the risk of developing pre-B acute lymphoblastic leukemia [12]. Negative factors can lead to non-stop division with the formation of immature, functionally defective cells with incorrectly formed B-cell receptors. At the stage of differentiation of pro-B cells, pre-B-I, and pre-B-II cells, the B-cell receptor is formed, which occurs due to the rearrangement of variable genes (V-genes) of immunoglobulins. The well-formed structure of B cell receptors determines the functional activity of B lymphocytes. 

The TIC compound stimulates an increase in the level of transitional B cells. The level of transitional B cells in the TIC group is higher than in the placebo, methyluracil, and untreated groups. Transitional B cells that differentiate into mature B1 cells and B2 cells, as well as B2 Cells, make up the bulk of the B lymphocyte population.

Transitional B cells differentiate into the main mature naïve B cells subpopulations: B1 cells, MZB, and B2 cells. We did not calculate the B1 cell subpopulation. B2 cells form the B-zone of primary FO-B cells in the lymph nodes and spleen. MZB lymphocytes are localized in the marginal zone of the spleen. MZB cells and FO-B cells are mitotically very active, have a huge variety of receptors, and respond to the stimulation of ligands and interleukins. At this stage, MZB cells and FO-B cells respond strongly to stimulation with the TIC compound, significantly exceeding the activity of the comparison drug methyluracil. The level of the TIC group reaches the value of the untreated group. At this stage, the quantitative sufficiency of MZB cells and FO-B cells is important. B2 cells and MZB cells express MHC II, CD80, and CD86 molecules, bind to the antigen, migrate to the T-dependent zone of the spleen, and present antigen to Th2. They form germinal center B cells in secondary follicles. MZB lymphocytes participate in the immune response to TD- and TI-antigens. Another characteristic feature of MZB lymphocytes is the expression of the CD1d molecules involved in the presentation of lipid antigens of the invariant natural killer T cells. Insufficiency of MZB cells and FO-B cells is quite often a predictor of an unfavorable prognosis of the course of infectious diseases, with the activation of opportunistic microflora. 

B2 cells and MZB cells bind to the antigen and differentiate into activated germinal center B cells. Activated germinal center B cells strongly respond to stimulation with the TIC compound, reaching the value of cells of untreated animals. The population cellularity of B cells in the germinal centers of the spleen is restored after cyclophosphamide-induced cytotoxic emptying. The activity of the TIC compound exceeds the activity of the comparison drug methyluracil by three times. Activated germinal center B cells can differentiate into memory B cells and plasma cells. 

We have not found compounds with a similar ability to stimulate the repair of MZB cells and FO-B cells in the spleen. 

Peptide preparations were used containing thymus hormones and thymus peptides: tactivin, thymosin, thymopoietin, thymostimulin, vilozen, thymomodulin, and timulin. These drugs stimulate the growth and proliferation of the T-lymphocytes and enhance the synthesis of tumor necrosis factor α and granulocyte–macrophage-colony-stimulating factor by monocytes [5,33]. Thymic drugs can indirectly affect the recovery of B-lymphocytic cells. However, thymic drugs stimulate the production of PGE2, activation of Th1 cells, proliferation of Th17 cells, and synthesis of IL-22 from Th22 cells via EP2 and EP4 receptors. All these factors lead to the development of chronic inflammatory processes with the development of autoimmune diseases **[40]**. 

The other group consists of drugs of bone marrow origin and contains the myelopeptides with a molecular weight of 500–3000 D produced by the bone marrow cells [5,8,33,34]. Myelopeptides include recombinant or purified humans, animal-colony-stimulating factors, and synthetic compounds. They include leukogen (a derivative of thiazolidindecarboxylic acid), Granocyte, filgrastim (human-colony-stimulating factor), lenograstim (recombinant human-glycosylated-colony-stimulating factor), pegfilgrastim (recombinant pegylated-human-colony-stimulating factor), empegfilgrastim, and myelopid (recombinant human-colony-stimulating factor). They stimulate leukopoiesis of granulocytic leukocytes, monocytes/macrophages, and the production of antibodies. They do not affect the recovery of the B-lymphocytic cell populations. Myelopeptides are a mixture of peptides and often cause a hyperthermic reaction, dizziness, and severe allergic reactions up to anaphylactic shock, nausea, vomiting, edema, sleep disturbance, and anorexia [5,8,9,10,11,33,34].

However, there are drugs that can stimulate the recovery of B-lymphoid cells. These are cytokine-based drugs (roncoleukin, betaleukin, romurtide) [9,10,12,13,39]. They are very effective drugs. They are used to stimulate interferogenesis, T- and B-lymphopoiesis [9], and immune reactions, and they participate in angiogenesis, apoptosis, chemotherapy, and embryogenesis [10,13]. However, cytokine drugs are very difficult to dose, and in minimal doses (10^−13^) can cause uncontrolled immune reactions up to a cytokine storm [12], septic shock, bradycardia, hypotension, vasodilation [13], tissue edema and microcirculation disorders in organs [14], blindness, and hypoglycemia [15]. They can be used only in a hospital under the supervision of a doctor.

Currently, herbal preparations are being developed to stimulate leukopoiesis. Some of these drugs are: Echinacea liquidum, Echinacea compositum C, Echinacea VILAR, Ginseng root, Siberian ginseng, pantocrine, and pantohematogen. A study of biologically active compounds that are part of plants is being conducted [16,17,18,38,39]. Currently, a myelo-stimulating compound obtained from the Ayurvedic plant Tinospora cordifolia [36] and from algae Chlorella vulgaris [18,37,38] is being developed and from the Dragon**’**s blood plant [41]. So far, these drugs are only adaptogens. 

There is one last group of drugs, which is practically not used because of its ability to stimulate the division of all cells, including bacterial and cancerous, and these are preparations based on nucleic acids. The use of oligonucleotides, such as ribozymes and antisense oligodeoxynucleotides (AS-ODN), has shown promise as therapeutic components, but has faced a number of problems, such as sensitivity to nucleases, side effects, and effective delivery [42].

## 4. Materials and Methods

### 4.1. Chemical Research Methods

#### Synthesis and Structure Studies for the Chemical Compound

The reaction progress and the purity of the compound were monitored using TLC on silica gel exposed to iodine vapor. “Nicolet 5700” FT-IR spectrometer (Thermo Fisher Scientific, Waltham, MA, United States) was used to record IR spectra. JNM-ECA “Jeol 400” spectrometer (Jeol, Tokio, Japan) was used to record 1 H (working frequency 399.78 MHz) and ^13^C (working frequency 100.53 MHz) NMR spectra with CDCl_3_ solvent. The residual protons (7.26 ppm) and carbon atom (77.0 ppm) signals of CDCl_3_ were used to determine chemical shifts [43]. The melting point is a criterion for the purity of a substance. A pure substance melts in a narrow temperature range from 0.1 to 1.0 C. The presence of impurities reduces the melting point of the substance and increases the interval between the beginning and end of melting. The complex 5-Benzyl-7-(o-fluorobenzylidene)-2,3-bis(o-fluorophenyl)-3,3a,4,5,6,7-hexahydro-2H-pyrazolo[4,3-c]pyridine has a m.p. of 145–146 °C, Rf 0.49 (Al_2_O_3_, eluent: benzene: dioxane = 10:1). A complex of 5-benzyl-7-(o-fluorobenzylidene)-2,3-bis(o-fluorophenyl)-3,3a,4,5,6,7-hexahydro-2H-pyrazolo[4,3-c]pyridine with β-cyclodextrin (natural polymer) was obtained in the form of a white powder melting at decomposition above 240 °C. Calculated for C_74_H_96_O_35_N_3_F_3_, %: C 54.02; H 5.84; N 2.55. Found, %: C 54.04; H 5.85; N 2.56. 

### 4.2. Biological Research Methods

#### 4.2.1. Experimental Animals

Mice C57BL6/J were acquired in the Charles River Laboratory (Hollister, California, USA) and propagated at the Kazakh Scientific Center for Quarantine and Zoonotic Infections named after Aikimbayev. Female laboratory mice 5–7 weeks of age weighing 16–20 g were used. Animals were bred and maintained in a pathogen-free barrier facility. 

#### 4.2.2. Design of Biological Experiment

Experiments were carried out in all groups: untreated (UT) group, TIC group, methyluracil (MU) group, and placebo (PL) group. Each group consisted of six mice. All mice C57BL6/J were kept at 21–23 °C, in standard polypropylene cages with free access to standard food and water. 

Four groups of mice: untreated, TIC, methyluracil, and placebo received cyclophosphamide cytostatics (Baxter Oncology GmbH, Halle (Westfalen, Germany). It was dissolved in saline solution. On the 1st, 3rd, and 5th day of the experiment, at 9:00 a.m., three injections were injected intramuscularly to three treated groups of mice: TIC group, methyluracil group, and placebo group in a dose of 100 mg/kg, in a volume of 0.05 mL (0.4% solution). Then, on the 8th, 10th, and 12th day of the experiment, at 9:00 a.m., the TIC group was injected with the compound trimecaine-based ionic compound (TIC). TIC compound was dissolved in saline solution and administered intramuscularly at a dose of 10 mg/kg, in a volume of 0.05 mL (0.4% solution). Methyluracil was injected with 6-methyluracil as a comparative compound. 

Methyluracil was dissolved in saline solution and administered intramuscularly at a dose of 10 mg/kg, in a volume of 0.05 mL (0.4% solution). Placebo group was injected with a saline solution (0.9% NaCl solution in a volume of 0.05 mL (0.4% solution)). Untreated group consisted of untreated animals (Figure 5). After the last injection, after 15 days of research work, the hind limb bones and spleen were surgically removed from anesthetized animals sacrificed by decapitation. The studies were performed in accordance with the “Rules for conducting preclinical (nonclinical) studies of biologically active substances” and “Ethical principles and recommendations for scientific experiments on animals” and the research was approved by the protocol of the Local Ethics Commission of the Al-Farabi Kazakh National University No. 20136/10, dated 12 June 2020. The validity period is 3 years [43,44,45].

#### 4.2.3. Preparation of Phosphate-Buffer Saline (PBS) and Column Buffer

PBS was used to wash the cells and prepare the column buffer solution. To prepare the PBS solution, NaH_2_PO_4_ and NaCl were dissolved in purified water, and NaOH was used to achieve pH 7.4. To prepare the column buffer, 0.5% FBS and 0.002 M EDTA were added to the PBS solution. All obtained solutions were passed through a sterile 0.22 µm membrane filter and stored in the refrigerator at 4–8 °C until use.

#### 4.2.4. Bone Marrow and Spleen Cellularity Assessment

Bone marrow and spleen were surgically removed from anesthetized animals sacrificed by decapitation. Bone marrow was washed with PBS solution using a 10 milliliter syringe. Contaminating erythrocytes were lysed with lysing solution (0.83% NH_4_Cl, 0.1% KHCO_3_, 0.003% EDTA, pH 7.2–7.4) for 10 min at room temperature, then cells were washed, filtered through 30 mL preparation filters (Miltenyi Biotec, San Jose, CA, USA), and re-suspended in PBS. Cells were counted and their viability was determined using trypan blue [43]. 

#### 4.2.5. Cytometric Studies of Bone Marrow and Spleen

The following mouse monoclonal antibodies (mAbs) were used for surface staining: APC-labeled anti-CD45R (B220), PE-labeled anti-CD43, Per CP-labeled anti-CD19, PE-labeled anti-MHCII, FITC-labeled anti-IgM, PE-labeled anti-CD40. Briefly, 10^6^ cells were incubated with mAbs specific for surface markers according to the manufacturer’s protocols and then fixed with fixation solution (cat. No 554722, BD Biosciences, Franklin Lakes, NJ, USA) for 20 min in dark at room temperature. Afterward, the cells were washed with PBS, re-suspended in flow solution, and immediately analyzed by flow cytometry on a FACSCalibur using CellQuest Pro software (BD Biosciences). Unstained cells, single fluorochrome-stained cells, and cells stained as fluorescence minus-one controls were used to set-up the flow cytometer. Multiparameter data were analyzed as described previously (Carleton and Nicholson, 2000). 

### 4.3. Statistic Data Processing

The data obtained were processed by mathematical statistics methods using Microsoft Excel (version 2210) and the “Statistica 6.0” software (version 6.0. The data are presented as an average value (M) ± standard deviation (SD) (*n* = 6 mice/group). We used one-way (single factor) analysis of variance (ANOVA) to determine a statistically significant difference and the values were considered reliable at *p* < 0.05 and F > F_crit_.

## 5. Conclusions

The TIC compound does not have a stimulating effect on increasing the level of pro-B cells, pre-B-I, and pre-B-II cells, which characterizes the TIC compound on the positive side. At the stage of pro- and pre-B lymphocytic differentiation, bone marrow cells should not be susceptible to physical and chemical factors that can lead to unlimited division of B cells with the production of immature functionally defective B cells.

The TIC compound stimulates the increase in the level of transitional B cells, which play an important role in the formation of a sufficient number of cell pools for further differentiation to mature B1 cells and mature B2 cells, where mature B2 cells make up the bulk of the B lymphocyte population.

MSB cells and FOB cells strongly react to stimulation by the TIC compound and reach the value of the untreated animals.

The TIC compound most strongly stimulates an increase in the volume of activated germinal center B cells, significantly exceeding the activity of the comparison drug methyluracil. Activated germinal center B cells completely restore population cellularity after cyclophosphamide-induced cytotoxic emptying of the follicular centers of the spleen.

Thus, the results of our experiments show that the TIC compound is more effective in B-lymphopoiesis-stimulating activity. 

The continued study of the pharmacodynamic and pharmacokinetic properties of the TIC compound will hopefully show the priority of the compound. The TIC compound does not contain nucleic acids and should not have the ability to stimulate the growth of bacterial mass and malignant cells. Usually, ionic liquids exhibit high bioavailability, low toxicity, and a long interval of action. If these properties are confirmed, then it will be possible to further develop this compound as a promising drug that stimulates B-lymphopoiesis.

## Figures and Tables

**Figure 1 molecules-28-01378-f001:**
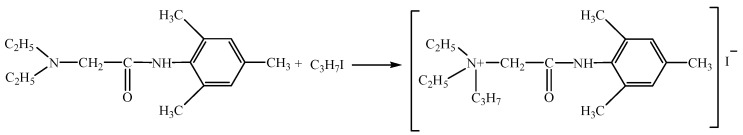
Chemical formation of the trimecaine-based ionic compound.

**Figure 2 molecules-28-01378-f002:**
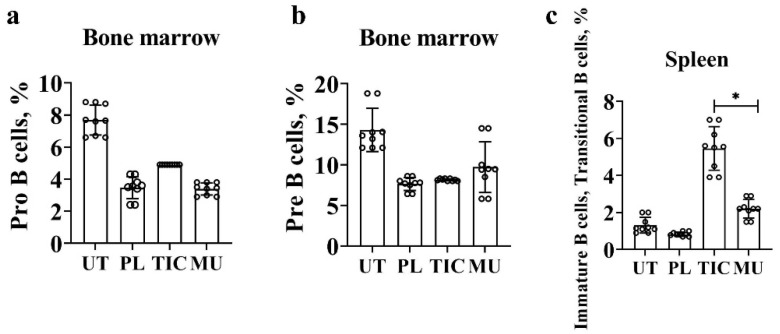
Results of flow cytometric phenotypic analysis. (**a**) The average level of pro-B cells (B220/CD45R^+^(APC)CD19^+^(PerCP)CD43^+^(PE)) in the bone marrow. (**b**) The average level of pre-B cells and immature B cells (B220/CD45R^+^(APC)CD19^+^(PerCP)CD43^−^(PE)) in the bone marrow. (**c**) The average level of immature B cells and transitional B cells (B220/CD45R+(APC)CD19^+^(PerCP)CD43^−^(PE)IgM^+^(FITS)) in the spleen. UT—untreated group, PL—placebo group, TIC—TIC group, MU—methyluracil group. The data are shown as mean value (M) ± standard deviation (SD) (*n* = 6 mice/group). * One-way analysis of variance (ANOVA) was used to determine a statistically significant difference and the values were considered reliable at *p* < 0.05 and F > F_crit_.

**Figure 3 molecules-28-01378-f003:**
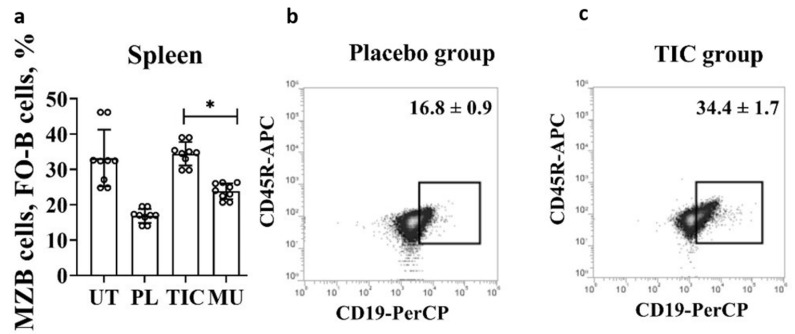
Results of flow cytometric phenotypic analysis of MZB cells and FO-B cells. (**a**) The average volume of MZB cells and FO-B cells (B220/CD45R^+^(APC)CD19^mid^(PerCP)CD43^−^(PE)) in the spleen. (**b**) The average volume of MZB cells and FO-B cells (B220/CD45R^+^(APC)CD19^mid^(PerCP)CD43^−^(PE)) in the placebo group. (**c**) The average volume of MZB cells and FO-B cells (B220/CD45R^+^(APC)CD19^mid^(PerCP)CD43^−^(PE)) in the TIC group. UT—untreated group, PL—placebo group, TIC—TIC group, MU—methyluracil group. The data are shown as mean value (M) ± standard deviation (SD) (*n* = 6 mice/group). * One-way ANOVA was used to determine a statistically significant difference and the values were considered reliable at *p* < 0.05 and F > F_crit_.

**Figure 4 molecules-28-01378-f004:**
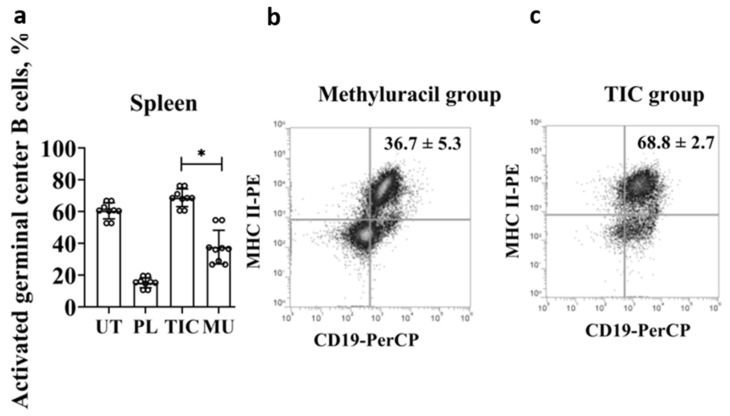
Results of flow cytometric phenotypic analysis of activated germinal center B cells. (**a**) The average level of activated germinal center B cells (B220/CD45R**^+^**(APC)CD19**^+^**(PerCP)CD40**^+^**(PE)MHCII**^+^**(PE)CD43**^+^**(PE)) in the spleen. (**b**) The average level of activated germinal center B cells (B220/CD45R**^+^**(APC)CD19**^+^**(PerCP)CD40**^+^**(PE)MHCII**^+^**(PE)CD43**^+^**(PE)) in the methyluracil group. (**c**) The average level of activated germinal center B cells (B220/CD45R**^+^**(APC)CD19**^+^**(PerCP)CD40**^+^**(PE)MHCII**^+^**(PE)CD43**^+^**(PE)) in the TIC group. UT—untreated group, PL—placebo group, TIC—TIC group, MU—methyluracil group. The data are shown as mean value (M) ± standard deviation (SD) (*n* = 6 mice/group). * One-way ANOVA was used to determine a statistically significant difference and the values were considered reliable at *p* < 0.05 and F > F_crit_.

**Figure 5 molecules-28-01378-f005:**
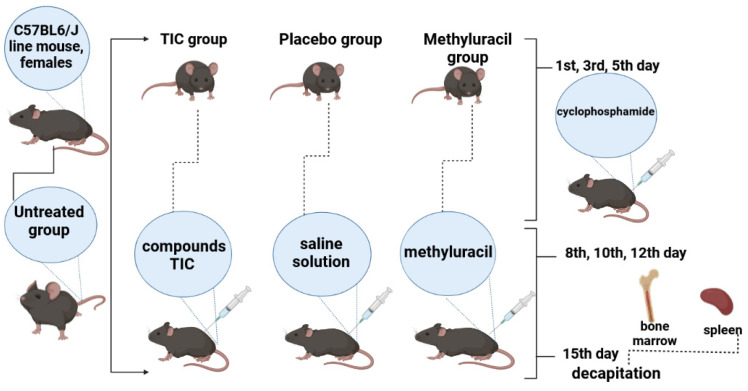
Experiment design.

**Table 1 molecules-28-01378-t001:** Bone marrow and spleen cellularity.

Group	Mean Number of NC Cells, ·10^6^ Cells	
*Bone Marrow*		
Untreated	25.1 ± 7.4	
Placebo	70.9 ± 7.2	P_TIC-PL_ < 0.01
		F > Fcrit
		32.36 > 10.12
TIC	17.2 ± 0.7 *	
Methyluracil	55.4 ± 7.1	P_TIC-MU_ < 0.02
		F > Fcrit
		17.29 > 10.12
*Spleen*		
Group	Mean organ weight, mg	Mean number of NC cells, ·10^6^ cells
Untreated	72.5 ± 17.0	92.1 ± 5.9
Placebo	60.0 ± 5.0	66.4 ± 5.9
TIC	45.0 ± 3.3	73.7 ± 5.4
Methyluracil	70.0 ± 5.0	60.6 ± 2.6

* *p* < 0.01 significance between Group TIC and Methyluracil group values.

## Data Availability

Not applicable.

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
