# Peer review of "Stimulation of B-Lymphopoiesis by Administration of a Trimecaine-Based Ionic Compound in Cyclophosphamide-Induced Hematopoietic-Depressive Model"

_molecules, 2023, doi:10.3390/molecules28031378_

Round 1

Reviewer 1 Report

The authors utilized TIC for stimulation of B-lymphopoiesis in this study. However, there may be major corrections and supplements necessary for the manuscript. Detailed comments and suggestion are listed below.

Major

1. The content of the abstract is insufficient to explain the limitations of the current drug. It would be nice if this paper could explain the limitations of the current drug and the improvement points of the technology in this paper more clearly.

2. The authors need to write a further description of the secondary B-lympholytic-depressive state in the introduction.

3. The risks of current medications are well documented. However, in the introduction, additional explanations are needed on the differences and improvements between the existing drugs and the technology of this paper.

4. Why did you choose Trimecaine? Authors should explain why they chose Trimecaine.

5. Materials and methods should have a clear title leading to the results. The materials and methods of this manuscript are not well understood for the purpose of the subheading.

6. n Figure 2, the authors need to add images of the composite structures and the resulting data to show the formation of the structures.

7. The description in the main text and the figure in the thesis should match. Four groups ('2.2.2. Intact, TIC, Control, Placebo.') are illustrated. However, Figure 1 shows only three groups.

9. Authors need corrections for Figure 3. In Figure 3, the distinction between uppercase and lowercase letters is confusing and even the subtitles are not uniform.

10. Did the authors measure body weight data in in vivo experiments?

12. In the conclusion section, the author should explain the advantages and expected effects of the technology presented in this paper.

Minor

1.     Overall, there are many parts where the letter spacing is not uniform. Correction is required.

2.     It is recommended to separate Figure 3 into two figures.

3.     Subheading and main body should be placed separately, and the first sentence of a paragraph should be indented.

4.     The % notation must be unified (line 116)

5.     The unit notation needs to be corrected. (line 139)

6.     Capital letters are used only at the beginning of a sentence or abbreviated words (line 235, line 240).

7.     It is recommended to write a discussion without a sub-heading.

8.     The reference form must be unified (line 349, line 361, line377-378, line 380-381).

Author Response

Major

  1. The content of the abstract is insufficient to explain the limitations of the current drug. It would be nice if this paper could explain the limitations of the current drug and the improvement points of the technology in this paper more clearly.

Answer: We added the explanation into the abstract section.

  1. The authors need to write a further description of the secondary B-lympholytic-depressive state in the introduction.

Answer: Added.

  1. The risks of current medications are well documented. However, in the introduction, additional explanations are needed on the differences and improvements between the existing drugs and the technology of this paper.

Answer: Added but in conclusion section

  1. Why did you choose Trimecaine? Authors should explain why they chose Trimecaine.

Answer: The justification on our choose is added to introduction section. Trimecaine was chosen by the authors for its unique properties (local anesthetic and antiarrhythmic agent) and non-toxicity, which are well known in medicine. At the same time, ionic compounds based on trimecaine have not been studied sufficiently, especially in terms of their pharmacological activity. The first reports on immunomodulating and plant growth regulating activities of the ionic compounds based on trimecaine were published in [21-31].

  1. Materials and methods should have a clear title leading to the results. The materials and methods of this manuscript are not well understood for the purpose of the subheading.

Answer: The subtitles were changed somewhere for better understanding

  1. Figure 2, the authors need to add images of the composite structures and the resulting data to show the formation of the structures.

Answer: Added the scheme of synthesis of ionic compound instead of just structure.

  1. The description in the main text and the figure in the thesis should match. Four groups ('2.2.2. Intact, TIC, Control, Placebo.') are illustrated. However, Figure 1 shows only three groups.

Answer: Figure I was changed

  1. Authors need corrections for Figure 3. In Figure 3, the distinction between uppercase and lowercase letters is confusing and even the subtitles are not uniform.

Answer: The figure 3 was divided into 2 figure to become more clear

  1. Did the authors measure body weight data in in vivo experiments?

Answer: The body weight of mice and spleen was measured in the control, intact, TIS group and the placebo group. The interval of body weight of mice in all groups was in the range of 18 - 22 g. No intensive weight gain and weight loss were observed. The difference with the body weight at the beginning of the experiment was 1.5 - 2 g. The weight of the spleen is shown in Table 1.

  1. In the conclusion section, the author should explain the advantages and expected effects of the technology presented in this paper.

 Answer: added

Minor

  1. Overall, there are many parts where the letter spacing is not uniform. Correction is required.

Answer: Done

  1. It is recommended to separate Figure 3 into two figures.

Answer: done

  1. Subheading and main body should be placed separately, and the first sentence of a paragraph should be indented.

Answer: Done

  1. The % notation must be unified (line 116)

Answer: Done.

  1. The unit notation needs to be corrected. (line 139)

Answer: Done.

  1. Capital letters are used only at the beginning of a sentence or abbreviated words (line 235, line 240).

Answer: Done

  1. It is recommended to write a discussion without a sub-heading.

Answer: Done

  1. The reference form must be unified (line 349, line 361, line377-378, line 380-381).

Answer: Done

Reviewer 2 Report

The manuscript entitled “Stimulation of B-lymphopoiesis in cyclophosphamide-induced hematopoietic-depressive model by administration of a Trimecaine-based ionic compound” describes a potential B cell-stimulating effect of a novel ionic compound. Although there are some possible interesting differences, the paper is very difficult to follow throughout. Since the methods and results are confusing, the significance of the study is not clear. Several issues should be addressed.

A major issue is that there are several grammatical errors, including many sentences that are fragments, which makes it difficult to understand. For instance, the title is confusing because it makes it sound like the depressive state is produced by the ionic compound. The first few sentences in the abstract are difficult to follow. Some things are provided as lists instead of sentences.

In the description of the study, it is not clear to what the “intact” group refers and there is no rationale provided for the use of 6-methyluracil as the control. Then in figure 1, the control group is distinct from the methyluracil group (even though the methyluracil was stated that it was used as a control in the description).

There are spelling errors in the figures (i.e., methyluracil is not spelled the same way in figure 1).

The labels on the figures are not clear – there are both upper case and lower case letters. The bars on the graphs should just be labeled as their group name instead of using the uppercase letters.

Minor

To what does “pericancrotic” refer?

Avoid the use of “etc” in manuscript since it is so vague.

Author Response

Comments and Suggestions for Authors 

  1. A major issue is that there are several grammatical errors, including many sentences that are fragments, which makes it difficult to understand. For instance, the title is confusing because it makes it sound like the depressive state is produced by the ionic compound. The first few sentences in the abstract are difficult to follow. Some things are provided as lists instead of sentences.

Answer: Changed the title to: Stimulation of B-lymphopoiesis by administration of a Trimecaine-based ionic compound in cyclophosphamide-induced hematopoietic-depressive model

In abstract:

acquired by them – deleted

The third sentence changed to:

The search for substances that stimulate B-lymphopoiesis among ionic compounds was motivated by the discovery of the unique properties of lidocaine docusate, one of the first ionic-liquid forms of the known drugs.

  1. In the description of the study, it is not clear to what the “intact” group refers and there is no rationale provided for the use of 6-methyluracil as the control. Then in figure 1, the control group is distinct from the methyluracil group (even though the methyluracil was stated that it was used as a control in the description).

 Answer: Added the explanation about the choice of methyluracil. 6-methyluracil is the most widely used drug in therapeutic practice in our country. It is an affordable and effective drug. Fig 1 changed

There are spelling errors in the figures (i.e., methyluracil is not spelled the same way in figure 1).

 Answer: Done, the figure 1 was changed

The labels on the figures are not clear – there are both upper case and lower case letters. The bars on the graphs should just be labeled as their group name instead of using the uppercase letters.

 Answer: Done

Minor

To what does “pericancrotic” refer?

Answer: Paracancrosis or pericancrosis is inflammation (sometimes purulent-destructive) around the tumor, which makes it difficult to treat neoplasms. Added the link https://medhelpsis.com/en/posts/12279

Avoid the use of “etc” in manuscript since it is so vagued

Answer: Done

Round 2

Reviewer 1 Report

All the reviewer's comments are addressed and satisfactory. Hence, may be accepted for publication in its present form.

Author Response

thank you!

Reviewer 2 Report

The manuscript entitled “Stimulation of B-lymphopoiesis by administration of a Trimecaine-based ionic compound in cyclophosphamide-induced hematopoietic-depressive model” was submitted as a revision. It is much improved; it reads well and the data are described better.  The title is much better as well. Figures are improved but there are still some areas of confusion as noted. There are also questions about the statistical analysis.

Minor suggestions:

Figure 3.2 f and g the protein for FITC should be on the axis. Also in the same figure for f and g, placebo and TIC were shown and for h and i, control and TIC were shown – was this the intention to show different groups in each of those figures?

It would be better to label the groups within the flow figures rather than only define them in the legend. I also think it would be easier to call the groups something “UT” for untreated instead of intact; “P” for placebo, “MU” for methyluracil instead of control (especially since some readers consider the placebo group a control) and “TIC” for the compound. It might also be better to use these kinds of labels so the reader is very clear on which group received cyclophosphamide (CP):  UT, CP + P, CP + MU, CP + TIC.

Why not just make figures 3.1 and 3.2 separate figures instead of continuing the lettering from the previous figure?

While the authors now provide some explanation for 6-methyluracil, they didn’t say WHY they used it as a control – yes it is widely used, but for what? Did they use it because they expected it to act similarly to TIC and increase B lymphopoiesis or because they expected it to inhibit and demonstrate an opposite effect of TIC? How does it serve as a control?

A student’s t-test is not the appropriate statistical test – a one-way ANOVA must be done since there are more than two groups.

Did the authors expect the huge increase in cells in response to cyclophosphamide in the bone marrow? Was this in response to the cellular depression in the spleen? This should be discussed.

The Discussion is still very results-heavy and needs more interpretation and comparison to other data in the field. Some of the new information provided in the conclusion should be better incorporated into the body of the discussion.

“Registration” is not the correct word in line 251 – perhaps “characterization” would be better

Author Response

Minor suggestions:

Figure 3.2 f and g the protein for FITC should be on the axis. Also in the same figure for f and g, placebo and TIC were shown and for h and i, control and TIC were shown – was this the intention to show different groups in each of those figures?

 Answer: The fig 3.2 and fig 3.1 were divided into three figures = fig 3, 4 and 5.

It would be better to label the groups within the flow figures rather than only define them in the legend. I also think it would be easier to call the groups something “UT” for untreated instead of intact; “P” for placebo, “MU” for methyluracil instead of control (especially since some readers consider the placebo group a control) and “TIC” for the compound. It might also be better to use these kinds of labels so the reader is very clear on which group received cyclophosphamide (CP):  UT, CP + P, CP + MU, CP + TIC.

 Answer: Done

Why not just make figures 3.1 and 3.2 separate figures instead of continuing the lettering from the previous figure?

 Answer: The figures were separated and the text was deleted from the figures.

While the authors now provide some explanation for 6-methyluracil, they didn’t say WHY they used it as a control – yes it is widely used, but for what? Did they use it because they expected it to act similarly to TIC and increase B lymphopoiesis or because they expected it to inhibit and demonstrate an opposite effect of TIC? How does it serve as a control?

 Answer: The first reason.

               Drugs that stimulate leukopoiesis used in clinical practice are represented by a spectrum: drugs of animal and human origin, synthetic drugs and herbal preparations.

               Sodium nucleinate was the first compound with leukopoiesis stimulating activity. Drugs of this series: Derinat, Poludan, Ridostin, Poludan, Inosine pranobex (isoprinosine), Methyluracil and Riboxin (Ballas et al., 1996; Gilkeson et al., 1998; Klinman et al., 1996). Methyluracil is the most widely used drug in this group.

               The following are peptide preparations containing thymus hormones: Thymosin, Thymopoietin and serum thymic factor Thymulin. These drugs stimulate the growth and proliferation of T-lymphocytes; enhance the synthesis of tumor necrosis factor α (TNFα) and granulocyte–macrophage colony-stimulating factor (GM-CSF) by monocytes (Azuma I., 1992; Ellouz et al., 1974; Murata et al., 1997; Novoseletskaya et al., 2015; Kato et al., 1985). Products containing thymus peptide: Thymalin, Thymopoietin, Thymostimulin, Vilozen, Thymomodulin. The drugs do not affect B-lymphopoiesis, only indirectly.

               Preparations of bone marrow origin are myelopeptides with a molecular weight of 500-3000 D, produced by bone marrow cells (Marzec et al., 1996; Fonina L.A. et al., 2012; Chereshnev V.A., 2012). These drugs are used to treat patients after chemotherapy and radiation therapy. Myelopeptides include recombinant or purified human colony-stimulating factors and synthetic drugs. Leucogen (a derivative of thiazolidindecarboxylic acid). It stimulates leukopoiesis of granulocytic leukocytes, monocytes/macrophages. The drug is contraindicated in Hodgkin's disease and leukemia. Granocyte (Lenograstim) is a recombinant human granulocyte colony stimulating factor. It also stimulates leukopoiesis of granulocytic leukocytes, monocytes/macrophages, activates phagocytosis and cytotoxicity of mature granulocytes. But myelopeptide drugs do not stimulate B-lymphopoiesis.  

               Cytokine preparations are structurally not similar to the TIС compound.

               Currently, herbal preparations are being developed to stimulate leukopoiesis. Some of these drugs are: Echinacea liquidum, Echinacea compositum C, Echinacea VILAR, ginseng root, Siberian ginseng, Pantocrine and Pantohematogen. A study of biologically active compounds that are part of plants is being conducted (Cetiner et al., 2005; Jahovic et al., 2003). Currently, a myelostimulating compound obtained from the Ayurvedic plant Tinospora cordifolia (Singh et al., 2006), from algae Chlorella vulgaris (de Souza Queiroz et al., 2004; Justo et al., 2001) is being developed (Ramos et al., 2010) and from the Dragon's blood plant (Ryan et al., 2016). But so far these drugs are only adaptogens.

               The second reason.

               The ionic compound N, N-dimethyl-N-(2-(methylamino)-2-oxoethyl) propane-1-ammonium iodide (TIC) is structurally more similar to methyluracil (2,4-dioxo-6-methylpyrimidine). Both compounds contain heterocyclic nuclei with a keto group in their chemical structures.

  1. Azuma I. Review: inducer of cytokines in vivo: overview of field and romurtide experience. Int J Immunopharmacol. 1992 Apr;14(3):487-96. doi: 10.1016/0192-0561(92)90180-s. PMID: 1618600.
  2. Ballas ZK, Rasmussen WL, Krieg AM. Induction of NK activity in murine and human cells by CpG motifs in oligodeoxynucleotides and bacterial DNA. J Immunol. 1996 Sep 1;157(5):1840-5. PMID: 8757300.
  3. Carras S, Valayer A, Moratal C, Weiss-Gayet M, Pages G, Morlé F, Mouchiroud G, Gobert S. Instructive role of M-CSF on commitment of bipotent myeloid cells involves ERK-dependent positive and negative signaling. J Leukoc Biol. 2016 Feb;99(2):311-9. doi: 10.1189/jlb.2A1214-619R. Epub 2015 Sep 2. PMID: 26336156.
  4. Cetiner M, Sener G, Sehirli AO, Ekşioğlu-Demiralp E, Ercan F, Sirvanci S, Gedik N, Akpulat S, Tecimer T, Yeğen BC. Taurine protects against methotrexate-induced toxicity and inhibits leukocyte death. Toxicol Appl Pharmacol. 2005 Nov 15;209(1):39-50. doi: 10.1016/j.taap.2005.03.009. PMID: 15890378.

5.      Chereshnev VA, Mazunina LS, Geĭn SV, Gavrilova TV, Chereshneva MV. [Effect of myelopeptides on reactive oxygen species generation and IL-1beta and TNF-alpha production by peripheral blood cells]. Patol Fiziol Eksp Ter. 2012 Jan-Mar;(1):19-22. Russian. PMID: 22629855.

6.      de Souza Queiroz J, Malacrida SA, Justo GZ, Queiroz ML. Myelopoietic response in mice exposed to acute cold/restraint stress: modulation by Chlorella vulgaris prophylactic treatment. Immunopharmacol Immunotoxicol. 2004 Aug;26(3):455-67. doi: 10.1081/iph-200026914. PMID: 15518178.

  1. Ellouz F, Adam A, Ciorbaru R, Lederer E. Minimal structural requirements for adjuvant activity of bacterial peptidoglycan derivatives. Biochem Biophys Res Commun. 1974 Aug 19;59(4):1317-25. doi: 10.1016/0006-291x(74)90458-6. PMID: 4606813.
  2. Fonina LA, Treshchalina EM, Belevskaia RG, Az'muko AA, Efremov MA, Sedakova LA, Kirilina EA. [Synthesis and anti-tumor properties of myelopeptide MP-1]. Bioorg Khim. 2012 Jul-Aug;38(4):406-12. doi: 10.1134/s1068162012040073. PMID: 23189554.
  3. Gilkeson GS, Conover J, Halpern M, Pisetsky DS, Feagin A, Klinman DM. Effects of bacterial DNA on cytokine production by (NZB/NZW)F1 mice. J Immunol. 1998 Oct 15;161(8):3890-5. PMID: 9780154.
  4. Jahovic N, Cevik H, Sehirli AO, Yeğen BC, Sener G. Melatonin prevents methotrexate-induced hepatorenal oxidative injury in rats. J Pineal Res. 2003 May;34(4):282-7. PMID: 12662351.
  5. Justo GZ, Silva MR, Queiroz ML. Effects of the green algae Chlorella vulgaris on the response of the host hematopoietic system to intraperitoneal ehrlich ascites tumor transplantation in mice. Immunopharmacol Immunotoxicol. 2001 Feb;23(1):119-32. doi: 10.1081/iph-100102573. PMID: 11322644.
  6. Kato K., Murota S. (1985) Lipoxygenase specific inhibitors inhibit murine lymphocyte reactivity to Con A by reducing IL-2 production and its action. Prostaglandins Leukot Med. 1985 Apr; 18(1):39-52.
  7. Klinman DM, Yi AK, Beaucage SL, Conover J, Krieg AM. CpG motifs present in bacteria DNA rapidly induce lymphocytes to secrete interleukin 6, interleukin 12, and interferon gamma. Proc Natl Acad Sci U S A. 1996 Apr 2;93(7):2879-83. doi: 10.1073/pnas.93.7.2879. PMID: 8610135; PMCID: PMC39727.
  8. Marzec E, Kubisz L, Jaroszyk F. Dielectric studies of proton transport in air-dried fully calcified and decalcified bone. Int J Biol Macromol. 1996 Feb;18(1-2):27-31. doi: 10.1016/0141-8130(95)01052-1. PMID: 8852750.
  9. Murata J, Kitamoto T, Ohya Y, Ouchi T. Effect of dimerization of the D-glucose analogue of muramyl dipeptide on stimulation of macrophage-like cells. Carbohydr Res. 1997 Jan 2;297(2):127-33. doi: 10.1016/s0008-6215(96)00261-3. PMID: 9060179.
  10. Novoseletskaya AV, Kiseleva NM, Zimina IV, Bystrova OV, Belova OV, Inozemtsev AN, Arion VY, Sergienko VI. Thymus Polypeptide Preparation Tactivin Restores Learning and Memory in Thymectomied Rats. Bull Exp Biol Med. 2015 Sep;159(5):623-5. doi: 10.1007/s10517-015-3030-6. Epub 2015 Oct 13. PMID: 26459479.
  11. Ramos AL, Torello CO, Queiroz ML. Chlorella vulgaris modulates immunomyelopoietic activity and enhances the resistance of tumor-bearing mice. Nutr Cancer. 2010;62(8):1170-80. doi: 10.1080/01635581.2010.513801. PMID: 21058206.

18.   Ran Y, Xu B, Wang R, Gao Q, Jia Q, Hasan M, Shan S, Ma H, Dai R, Deng Y, Qing H. Dragon's blood extracts reduce radiation-induced peripheral blood injury and protects human megakaryocyte cells from GM-CSF withdraw-induced apoptosis. Phys Med. 2016 Jan;32(1):84-93. doi: 10.1016/j.ejmp.2015.09.010. PMID: 26527395.

19.   Singh SM, Singh N, Shrivastava P. Effect of alcoholic extract of Ayurvedic herb Tinospora cordifolia on the proliferation and myeloid differentiation of bone marrow precursor cells in a tumor-bearing host. Fitoterapia. 2006 Jan;77(1):1-11. doi: 10.1016/j.fitote.2005.05.002. Epub 2005 Dec 2. PMID: 16326030.

  1. Taki M, Tsuboi I, Harada T, Naito M, Hara H, Inoue T, Aizawa S. Lipopolysaccharide reciprocally alters the stromal cell-regulated positive and negative balance between myelopoiesis and B lymphopoiesis in C57BL/6 mice. Biol Pharm Bull. 2014;37(12):1872-81. doi: 10.1248/bpb.b14-00279. PMID: 25451836.
  2. Vartapetian AB. Belki myc, protimozin alpha i kletochnoe delenie [Myc proteins, prothymosin alpha, and cell division]. 1992 Mar;57(3):477-8. Russian. PMID: 1344198.

A student’s t-test is not the appropriate statistical test – a one-way ANOVA must be done since there are more than two groups.

 Answer: We did and added ANOVA test into the article.

Did the authors expect the huge increase in cells in response to cyclophosphamide in the bone marrow? Was this in response to the cellular depression in the spleen? This should be discussed.

 Answer: this part added into discussion section

The Discussion is still very results-heavy and needs more interpretation and comparison to other data in the field. Some of the new information provided in the conclusion should be better incorporated into the body of the discussion.

 Answer: discussion was expanded.

“Registration” is not the correct word in line 251 – perhaps “characterization” would be better

Answer: Done

Round 3

Reviewer 2 Report

I appreciate the detailed rationale for methyluracil but now it is so much information that it is difficult to read as presented. I only intended for the authors to provide a sentence or two with a rationale of why methyluracil is considered a control. 

Thank you for changing group names - it is much improved. In section 5 last sentence there is still an "intact" there.

There are almost no references in the discussion, which means that the authors have still not discussed their results as compared to other literature. Actually, the information that was provided on the justification for methyluracil could be PART of the discussion if they want to move it there. 

It is not clear to what the "BIV compound" refers in the new part of the discussion.

Author Response

We appreciate reviewers and editors for their work. Authors responded to all remarks and ask to change the grants numbers in Funding section.